# Viral-inducible Argonaute18 confers broad-spectrum virus resistance in rice by sequestering a host microRNA

Jianguo Wu[1,2†], Zhirui Yang[1†], Yu Wang[1], Lijia Zheng[1,2], Ruiqiang Ye[3], Yinghua Ji[4], Shanshan Zhao[1], Shaoyi Ji[1], Ruofei Liu[1], Le Xu[3], Hong Zheng[1], Yijun Zhou[4], Xin Zhang[5], Xiaofeng Cao[6], Lianhui Xie[2], Zujian Wu[2]*, Yijun Qi[3]*, Yi Li[1]*

[1]State Key Laboratory of Protein and Plant Gene Research, College of Life Sciences, Peking University, Beijing, China; [2]Fujian Province Key Laboratory of Plant Virology, Institute of Plant Virology, Fujian Agriculture and Forestry University, Fuzhou, China; [3]Center for Plant Biology, Tsinghua-Peking Center for Life Sciences, College of Life Sciences, Tsinghua University, Beijing, China; [4]Institute of Plant Protection, Jiangsu Academy of Agricultural Sciences, Nanjing, China; [5]Institute of Crop Science, Chinese Academy of Agricultural Sciences, Beijing, China; [6]State Key Laboratory of Plant Genomics and National Center for Plant Gene Research, Institute of Genetics and Developmental Biology, Beijing, China

**Abstract** Viral pathogens are a major threat to rice production worldwide. Although RNA interference (RNAi) is known to mediate antiviral immunity in plant and animal models, the mechanism of antiviral RNAi in rice and other economically important crops is poorly understood. Here, we report that rice resistance to evolutionarily diverse viruses requires Argonaute18 (AGO18). Genetic studies reveal that the antiviral function of AGO18 depends on its activity to sequester microRNA168 (miR168) to alleviate repression of rice AGO1 essential for antiviral RNAi. Expression of miR168-resistant *AGO1a* in *ago18* background rescues or increases rice antiviral activity. Notably, stable transgenic expression of AGO18 confers broad-spectrum virus resistance in rice. Our findings uncover a novel cooperative antiviral activity of two distinct AGO proteins and suggest a new strategy for the control of viral diseases in rice.

*For correspondence: wuzujian@ 126.com (ZW); qiyijun@tsinghua. edu.cn (YQ); liyi@pku.edu.cn (YL)

†These authors contributed equally to this work

**Competing interests:** The authors declare that no competing interests exist.

## Introduction

Small RNAs including small interfering RNAs (siRNAs) and microRNAs (miRNAs) are important regulators of gene expression via RNA interference (RNAi) pathways in many organisms (*Baulcombe, 2004*; *Wang et al., 2011*; *Hauptmann and Meister, 2013*; *Meister, 2013*). The RNAi machinery contains many proteins involved in the biogenesis and function of specific types of small RNAs (*Chen, 2009*; *Wang et al., 2011*; *Hauptmann and Meister, 2013*; *Meister, 2013*). Among these are two categories that act as the core of the RNA silencing machinery: DICER (DICER-LIKE or DCL in plants) and ARGONAUTE (AGO). DICER and DCL proteins mainly function to process double-stranded or folded stem-loop precursor RNAs into small RNA duplexes. AGO proteins incorporate one strand of an siRNA duplex to form an effector complex called RNA-induced silencing complex (RISC) to regulate the stability/translatability of target RNAs or to mediate methylation of target DNA sequences (*Baulcombe, 2004*; *Wang et al., 2011*; *Meister, 2013*; *Pfaff et al., 2013*; *Li et al., 2013a*; *Rogers and Chen, 2013*).

**eLife digest** Rice is a major food crop, providing over a fifth of all calories consumed by people around the world. As such, it is important to find ways to prevent the diseases that affect rice plants. Many of the viruses that infect rice are transferred between plants by insects and many insects carry more than one virus at a time; this means it can be difficult to predict where a disease will next emerge. As a result, there is a pressing need to develop new and effective strategies that boost the ability of rice plants to fight off harmful viruses.

One way that plants defend themselves from viruses involves using a system called RNA interference to identify and destroy the RNA molecules that viruses produce. This process depends on the Argonaute (AGO) family of proteins, although the roles of many of its members are not well understood. One of the better-studied AGO proteins is called AGO1 and is known to be important for defending plants against viruses. Unfortunately, a small RNA molecule called miR168 acts to limit the amount of AGO1 in a cell, and the levels of miR168 increase in virus-infected rice plants.

Wu, Yang et al. exposed rice plants to two species of insect that each carried a different plant virus. Rice plants infected with these viruses increased their levels of both AGO1 and another AGO protein called AGO18. Modifying the ability of rice plants to produce AGO18 revealed that the anti-viral activity of AGO1 is abolished in plants lacking AGO18. However, plants that over-produce AGO18 are better able to fight off viral infections. Wu, Yang et al. further showed that AGO18 binds to miR168 and so prevents this small RNA from reducing AGO1 levels. Therefore, AGO1 and AGO18 work together to defend rice plants from viruses.

Wu, Yang et al. suggest that engineering rice plants to make more AGO18 could make them more resistant to viruses. Further work will be needed to confirm whether AGO1 and AGO18 also work together to defend rice against viruses other than the two tested so far and to investigate whether these proteins also perform similar roles in other crops.

In plants, the RNAi machinery has undergone extensive amplification and functional specialization during evolution. For instance, the model plant *Arabidopsis thaliana* encodes four DCLs that function indistinct and yet overlapping RNAi pathways to control diverse biological processes ranging from development, response to abiotic stresses, to defense against pathogens (*Deleris et al., 2006*; *Chapman and Carrington, 2007*; *Garcia-Ruiz et al., 2010*; *Vazquez et al., 2010*). *Arabidopsis* encodes 10 AGOs whose functions are not all understood. Well-studied AGOs include AGO1 that mediates mRNA cleavage is critical for development, AGO4 that directs DNA methylation, and AGO2 that functions in DNA double strand break repair (*Baumberger and Baulcombe, 2005*; *Mallory and Vaucheret, 2010*; *Ye et al., 2012*; *Wei et al., 2012*). AGO1 is particularly notable in that its homeostasis is controlled at the transcriptional, post-transcriptional, and post-translational levels. At the post-transcriptional level, the *AGO1* mRNA is a target of miR168. Therefore, miR168-guided cleavage of *AGO1* mRNA by AGO1 protein exerts auto-regulation. Moreover, *AGO1* is co-expressed with miR168 and AGO1 protein can stabilize miR168 post-transcriptionally (*Vaucheret et al., 2006*; *Vaucheret, 2008*; *Mallory and Vaucheret, 2010*). At the post-translational level, the accumulation of AGO1 can be reduced by F-box proteins in a proteasome-independent manner through the autophagy pathway (*Derrien et al., 2012*; *Rogers and Chen, 2013*). The auto-regulation of AGO1 indicates that its level within a cell can be dynamic and this dynamics may significantly impact the biological activities of a plant.

RNA-mediated immunity against viruses operates in plants, fungi, invertebrates, and mammals to specifically destroy viral RNAs through the cellular RNA silencing machinery (*Li et al., 2013b*; *Maillard et al., 2013*). In plants, it is well known that AGO1 is a major effector of antiviral RNAi; AGO1 associates with virus-derived siRNAs (vsiRNAs) and mediates the degradation of viral RNAs. Furthermore, AGO2 and AGO7 are induced during viral infection, and both proteins can bind viral siRNAs. The antiviral function of AGO2 and AGO7 requires their slicing activity (*Qu et al., 2008*; *Wang et al., 2011*). AGO2 is repressed by AGO1-associated miR403, and AGO1 and AGO2 appear to exert antiviral functions in a non-redundant and cooperative manner. Specifically, AGO1 functions in the first layer of antiviral RNAi; when AGO1's antiviral function is inhibited, a second layer is activated involving AGO2 (*Harvey et al., 2011*; *Jaubert et al., 2011*; *Scholthof et al., 2011*;

*Wang et al., 2011*; *Carbonell et al., 2012*; *Xia et al., 2014*). AGO2 also recruits miR393* to regulate plant immunity against bacterial infection (*Zhang et al., 2011*). As a counter-defense strategy, some plant viruses have evolved silencing suppressors to target AGO1 (*Burgyán and Havelda, 2011*). Moreover, infection of many viruses can elevate the miR168 level to down-regulate AGO1, thereby nullifying this layer of host defense (*Várallyay et al., 2010*). Thus, regulation of AGO1 by both host and viral factors plays a critical role in determining host responses to viral infection. Whether a host has positive regulators to check the viral counter-defense activities is not understood. How different AGOs have evolved to regulate plant responses to pathogen infection also remains an outstanding question (*Ding and Voinnet, 2007*; *Ding, 2010*; *Garcia-Ruiz et al., 2010*).

Rice (*Oryza sativa*) is one of the most important food crops as well as an experimental model plant for monocotyledonous plants (monocots). However, rice production and consequently food sustainability is under the constant threat of emerging and reemerging viral diseases. Rice viral pathogens are genetically diverse and many highly pathogenic viruses such as *Rice stripe Tenuivirus* (RSV, with a genome comprising 4 negative-stranded RNAs) and *Rice dwarf Phytoreovirus* (RDV, with a genome comprising 12 double-stranded RNAs) are transmitted persistently and solely by arthropod vectors (*Hibino, 1996*; *Ren et al., 2010*; *Du et al., 2011*). Because of the global circulation of these vectors and lack of virus resistance germplasms, the incidence and severity of rice viral diseases in many rice-growing regions are unpredictable. Infection by multiple viruses is also a common and severe challenge for other important crops. Therefore, developing new and effective strategies to control infection by multiple viruses for a crop, especially the prevalent food crops in the monocot group such as rice, maize, and wheat that have traditionally been under-investigated, is of paramount importance for human food sustainability. Successful development of such strategies requires a knowledge base of the mechanisms of viral infection and host defense responses.

The rice genome encodes five DCLs and 19 AGOs. How different AGOs regulate antiviral RNAi in rice is not known. In this study, we report a critical role of RNAi in antiviral defense in rice under natural infection conditions, when the plants were inoculated, as in the field, by viruliferous insect vectors brown planthopper (*Laodelphax striatellus*) and leafhopper (*Nephotettix cincticeps*) that transmit RSV and RDV, respectively. Intriguingly, we found that AGO18, a member of a new AGO clade that is conserved in monocots, is specifically induced by the infection of two taxonomically different viruses and is required for the antiviral function of AGO1. Loss-of-function *ago18* mutation abolishes, whereas over-expression of AGO18 increases, the AGO1 antiviral activity. We further demonstrated that AGO18 competes with AGO1 for binding miR168, resulting in elevated levels of AGO1 in the infected plants to enable antiviral defense. Expression of an miR168-resistant *AGO1a* variant in the *ago18* background rescues or increases rice antiviral activity. The antiviral function of AGO18 requires its small RNA-binding, not slicing activity. Our findings reveal a novel mechanism AGO1 homeostasis regulation by AGO18 for antiviral defense and have significant implications in understanding the evolutionary amplification of RNA silencing mechanisms and in developing novel antiviral strategies.

## Results

### A role for AGO1 in rice antiviral immunity under natural infection conditions

Our understanding of RNAi as an antiviral defense mechanism in plants mainly comes from viral infection systems involving model plants *Arabidopsis* and *Nicotiana benthamiana* by mechanical inoculation with virions or in vitro viral RNA transcripts or by agro-infiltration with viral cDNAs. To investigate the importance of RNAi in antiviral immunity in economically important crops under natural infection conditions, we used the rice-RSV patho-system in which rice is infected by RSV through the transmission of insect vector brown planthopper.

Given the established role of AGO1 in antiviral defense in *Arabidopsis*, we tested whether AGO1 functions similarly in rice by inoculating an *ago1* RNAi line (*Wu et al., 2009*) by viruliferous brown planthopper carrying RSV to recapitulate the natural infection conditions. As shown in *Figure 1A*, the a*go1* RNAi line, in which the expression of *AGO1s* is diminished (*Figure 1—figure supplement 1A*), was much more susceptible to RSV infection and showed more severe symptoms. Northern blots showed a remarkable increase in the accumulation of RSV genomic RNAs in the *ago1* RNAi line (*Figure 1B*). Since *AGO1a* and *1b* are expressed at much higher levels than *AGO1c* and *1d* in both mock-inoculated and RSV-infected rice plants (*Figure 1—figure supplement 1B,C*)

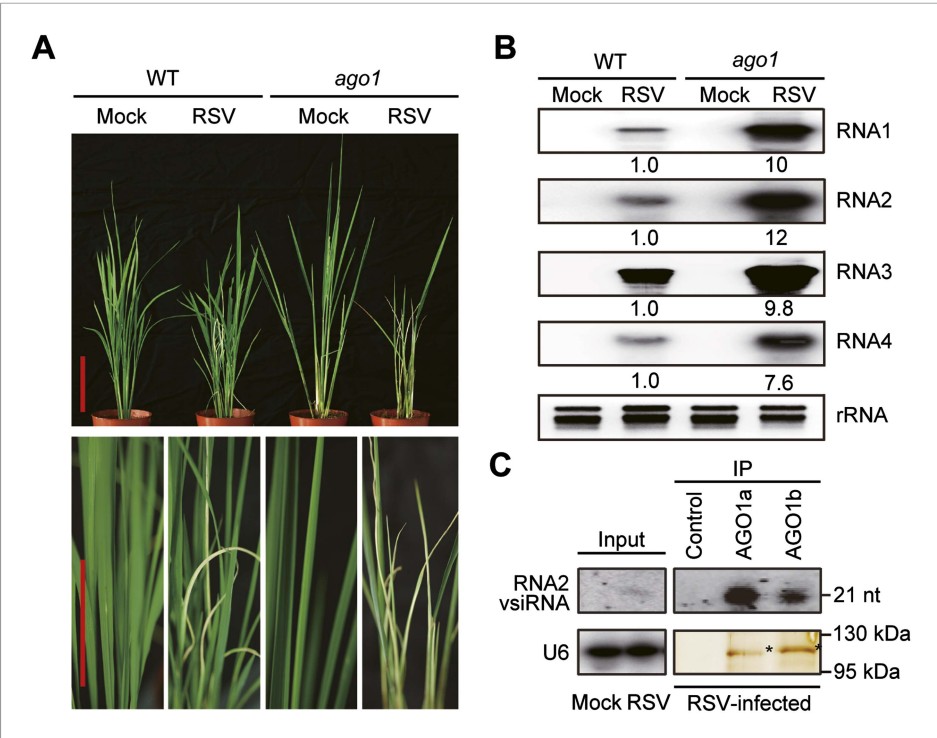

**Figure 1**. AGO1 participates in antiviral immunity in rice. (**A**) Symptoms of WT and *ago1* RNAi rice plants infected with RSV, pictures were taken at 6 weeks post-inoculation. Scale bars, 15 cm. (**B**) Detection of RSV genomic RNA segments in WT and *ago1* RNAi rice plants by Northern blot. The blots were hybridized with radiolabeled riboprobes specific for each RNA segment. rRNAs were stained with ethidium bromide and served as loading controls. The RNA signals were quantified and normalized to rRNAs, and the relative values were calculated by comparison with those in RSV-infected WT (arbitrarily set to 1.0). (**C**) Detection of vsiRNAs associated with AGO1a and AGO1b immunoprecipitates. The silver-stained gel shows that comparable amounts of AGO1 complexes were used for RNA preparation. The asterisks indicate the positions of AGO proteins. The position of a RNA size marker, electrophoresed in parallel, is shown to the right of the blots. U6 was also probed and served as a loading control.

The following figure supplement is available for figure 1:

**Figure supplement 1**. Characterization of rice *ago1* RNAi lines and expression profiles of rice *AGO1a*, *1b*, *1c*, and *1d*.

(**Kapoor et al., 2008**), we presumed that AGO1a and 1b are more important for antiviral defense. Therefore, AGO1a and 1b were characterized further in subsequent studies. We immunoprecipitated AGO1a and 1b complexes from RSV-infected or mock-treated rice plants using specific antibodies. Northern blot analyses showed that RSV vsiRNAs were readily detected in the AGO1a and AGO1b complexes (**Figure 1C**), suggesting that AGO1 is an effector of vsiRNA. Taking together, these data indicate a role for AGO1s in antiviral immunity under natural infection conditions in rice.

## AGO18 is specifically induced by viral infection and required for antiviral immunity

We have previously shown that rice *AGO1* as well as *AGO18*, a member of a new AGO clade that is conserved in monocots (**Figure 2—figure supplement 1**), is highly induced by viral infection (**Du et al., 2011**). Consistent with mRNA expression patterns, AGO18 protein was hardly detectable in mock-inoculated rice plants but accumulated to a high level in RSV-infected plants (**Figure 2A**). To further test whether RSV infection induces the transcription activity of *AGO18* promoter, we generated transgenic rice plants that express β-glucuronidase (GUS) under the control of *AGO18* promoter. RSV infection led to high levels of *GUS* expression in such plants as compared to the

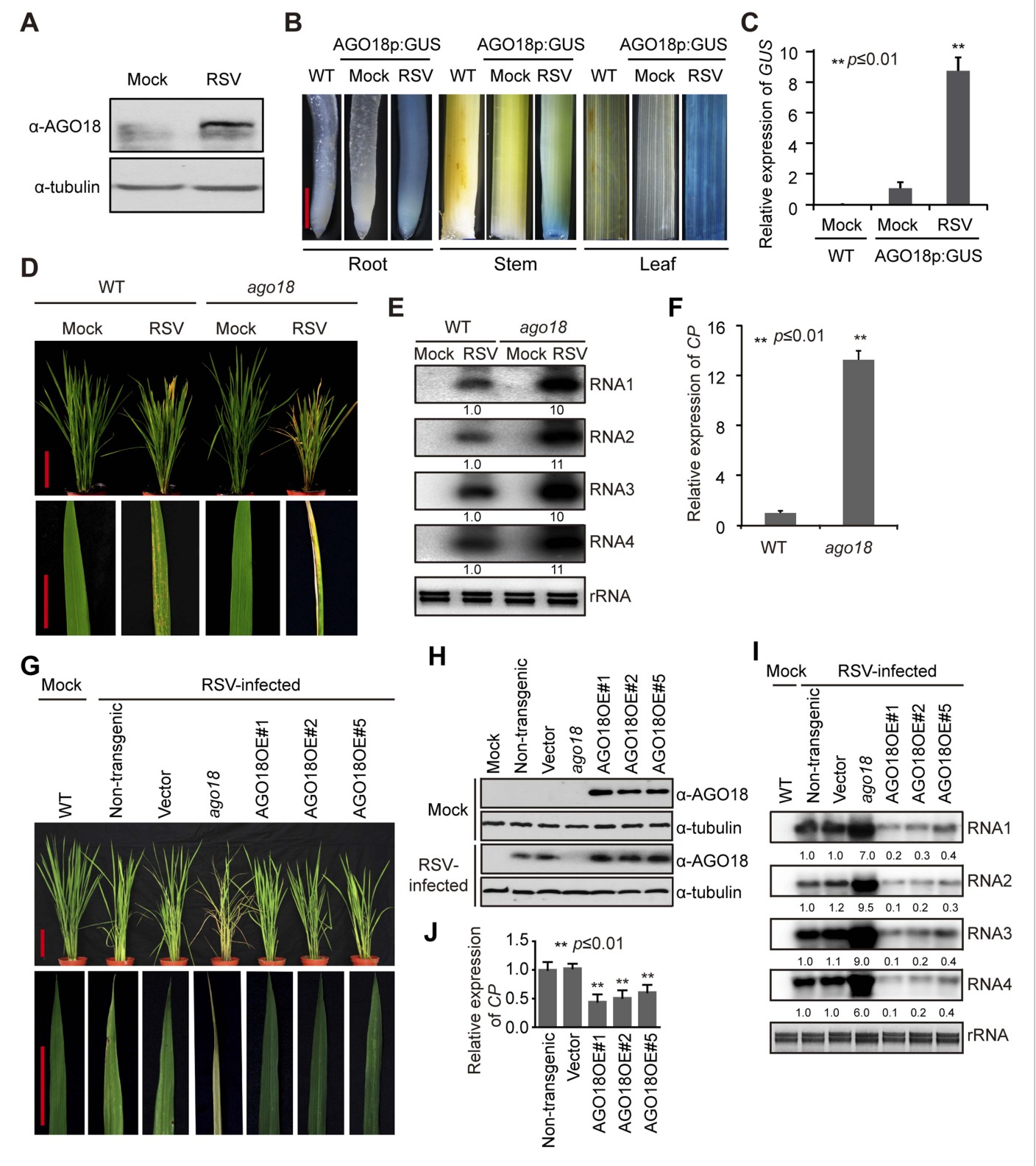

**Figure 2**. AGO18 is induced by viral infection and confers antiviral immunity in rice. (**A**) Detection of AGO18 in mock- or RSV-inoculated rice plants by Western blot. Tubulin was probed and served as a loading control. (**B**) Representative GUS staining images of mock- or RSV-inoculated *AGO18p:GUS* transgenic plants. Scale bars, 5 mm. (**C**) qRT-PCR analysis of the transcript level of *GUS* in the indicated plants. The average (± standard deviation) values from three biological repeats of qRT-PCR are shown. (**D**) Symptoms of wild-type (WT) and *ago18* rice plants infected with RSV, pictures were taken at 6

*Figure 2. continued on next page*

Figure 2. Continued

weeks post-inoculation. Scale bars, 15 cm. (**E**) Detection of RSV genomic RNA segments in WT and *ago18* rice plants by Northern blot. The blots were hybridized with radiolabeled riboprobes specific for each RNA segment. rRNAs were stained with ethidium bromide and served as loading controls. The RNA signals were quantified and normalized to rRNAs, and the relative values were calculated by comparison with those in RSV-infected WT (arbitrarily set to 1.0). (**F**) Detection of RSV *CP* in the RSV-infected WT and *ago18* by qRT-PCR. The expression levels were normalized using the signal from *OsEF-1a*. The average (± standard deviation) values from three biological repeats of qRT-PCR are shown. (**G**) Symptoms of RSV-infected WT (Non-transgenic), *ago18* and transgenic lines overexpressing AGO18, pictures were taken at 6 weeks post-inoculation. Scale bars, 15 cm (upper panel) and 5 cm (lower panel). (**H**) Detection of AGO18 in mock- or RSV-inoculated plants as indicated. Tubulin was probed and served as a loading control. (**I**) Detection of RSV genomic RNA segments in the indicated plants by Northern blot. The blots were hybridized with radiolabeled riboprobes specific for each RNA segment. rRNAs were stained with ethidium bromide and served as loading controls. The RNA signals were quantified and normalized to rRNAs, and the relative values were calculated by comparison with those in RSV-infected WT (Non-transgenic) (arbitrarily set to 1.0). (**J**) Detection of RSV *CP* in the indicated RSV-infected plants by qRT-PCR. The expression levels were normalized using the signal from *OsEF-1a*. The average (± standard deviation) values from three biological repeats of qRT-PCR are shown.

The following figure supplements are available for figure 2:

**Figure supplement 1**. Phylogenetic analysis of AGO family proteins in plants.

**Figure supplement 2**. RSV CP triggers AGO18 expression.

**Figure supplement 3**. Identification of the *ago18* mutant.

**Figure supplement 4**. AGO18 is required for rice resistance to RDV infection.

background levels in mock-inoculated plants (*Figure 2B,C*). These data demonstrate that RSV infection, but not insect vector feeding, induced *AGO18* expression at the transcriptional level.

To investigate how RSV infection triggers the expression of *AGO18*, we examined the possibility that viral proteins produced during infection induce AGO18 expression. To do this, we generated transgenic rice plants that over-express Myc-tagged RSV P2, CP, pC4, and P4 proteins (Xiong et al., 2008), respectively (*Figure 2—figure supplement 2*). We found that AGO18 was induced only in the transgenic lines that over-expressed CP, but not in those that over-expressed three other viral proteins (*Figure 2—figure supplement 2*). This suggests that RSV CP is an effector to induce AGO18 expression during viral infection.

The induction of AGO18 expression by RSV infection prompted us to investigate its function in antiviral defense. We obtained a *Tos17* retrotransposon insertional mutant *ago18* (NF6013) (*Figure 2—figure supplement 3A*) from the *Tos17* database and isolated homozygous plants (*Figure 2—figure supplement 3B*). The *ago18* mutant did not exhibit notable phenotypic differences from the WT plants in growth and development but were much more sensitive to RSV infection than the WT plants (*Figure 2D*). Viral replication and infection rates also increased in *ago18* mutant plants (*Figure 2E,F*, and *Supplementary file 1A*). These results suggest that AGO18 may be dispensable for normal growth and development of rice, but is required for defense against RSV infection. Importantly, AGO1 is present in the *ago18* mutants. Therefore, these data indicate that the antiviral function of AGO1 depends on the presence of AGO18. We further over-expressed *AGO18* under the control of the *ACTIN* promoter in the WT rice background. The antiviral activity of AGO18 in the three independent transgenic lines was directly correlated with the expression levels of AGO18 (*Figure 2G–J*).

AGO18 was not only induced by and functioned against RSV, but also was induced by and functioned against RDV, a double-stranded RNA *Phytoreovirus* (*Figure 2—figure supplement 4*). This indicates that AGO18 is effective in positively regulating defense against a broad spectrum of viruses with diverse genome structures.

## AGO18 unlikely functions as an effector of vsiRNAs

Given that AGO1 is present in the *ago18* mutant and AGO18 is present in *ago1* RNAi plant and that both *ago18* mutant and *ago1* RNAi rice plants were very susceptible to viral infection, AGO1 and AGO18 evidently depend on each other for their antiviral activities. A key question is how this mutual dependence operates.

To investigate how AGO18 acts in antiviral defense, we first tested whether this protein may function as an effector of vsiRNAs, like AGO1. We immunoprecipitated AGO18 complexes from RSV-infected WT rice plants using AGO18 antibodies (*Figure 3A*). Northern blots using RSV RNA2 probes showed much weaker signals for vsiRNAs in AGO18 (*Figure 3A*), compared to those in AGO1a and AGO1b (*Figure 1C*). We further profiled small RNAs in AGO18 complexes prepared from RSV-infected plants using deep sequencing. For comparison, small RNAs in total extracts, AGO1a, and AGO1b complexes were also profiled in parallel. We found that vsiRNAs accounted for about 4.48% of the total small RNAs in RSV-infected plants and 17.37%–9.77% of those in the AGO1a and AGO1b complexes (*Figure 3B* and *Supplementary file 1B*), indicating that vsiRNAs are highly enriched in the AGO1 complexes. In contrast, vsiRNAs accounted for only about 2.52% of the AGO18-bound small RNAs (*Figure 3B* and *Supplementary file 1B*). These data suggest that AGO18 is not a major effector of vsiRNAs. Together with the finding that AGO18 is required for the antiviral activity of AGO1 (see above), this observation suggests that AGO18 very likely uses a different strategy to regulate antiviral defense response in rice.

## AGO18 competes with AGO1 for miR168 to up-regulate AGO1 upon viral infection

RSV infection increases the accumulation of miR168 as well as AGO18 (*Du et al., 2011*) (*Figure 2A–C* and *Figure 4A*), suggesting that the increased miR168 is not efficiently loaded into AGO1 and hence not effective in targeting AGO1. Intriguingly, from the small RNA deep sequencing analyses, we found that AGO18 recruited a large amount of miR168 in RSV-infected rice plants compared with AGO1a or AGO1b (*Figure 4A* and *Supplementary file 1C*). Given that miR168 plays a critical role in AGO1 homeostasis in plants (*Mallory and Vaucheret, 2010*) and that the antiviral function of AGO1 requires the presence of AGO18, our observations suggest that AGO18 up-regulates AGO1 by competitively binding miR168 to enable antiviral defense.

Immunoprecipitation (IP)-northern blot analyses further confirmed that RSV infection of WT rice plants increased the association of miR168 with AGO18 but decreased its association with AGO1a and 1b, whereas several control miR-NAs (including miR166 and miR156) showed no obvious changes (*Figure 4B*). Consistent with these results, in RSV-infected *ago18* mutants, where AGO18 is absent, more miR168 was now loaded into AGO1a and 1b (*Figure 4C*), which was correlated with the reduced expression of *AGO1* at mRNA (*Figure 4D*) and protein (*Figure 4E*) levels. Also in the *AGO18OE#1* transgenic rice line, in which *AGO18* was overexpressed, *AGO1a* and *1b* both increased at the mRNA level (*Figure 4F*), likely as a result of more miR168 being loaded into AGO18. Consistent with our previous finding (*Du et al., 2011*), *AGO2* was induced by RSV infection in wild-type rice plants (*Figure 4F*). Intriguingly, knockout or overexpression of AGO18 did not have an obvious effect on such induction, indicating that the induction of *AGO2* by RSV infection is independent of AGO18 (*Figure 4F*). Thus, AGO18 specifically up-regulates AGO1 in antiviral defense.

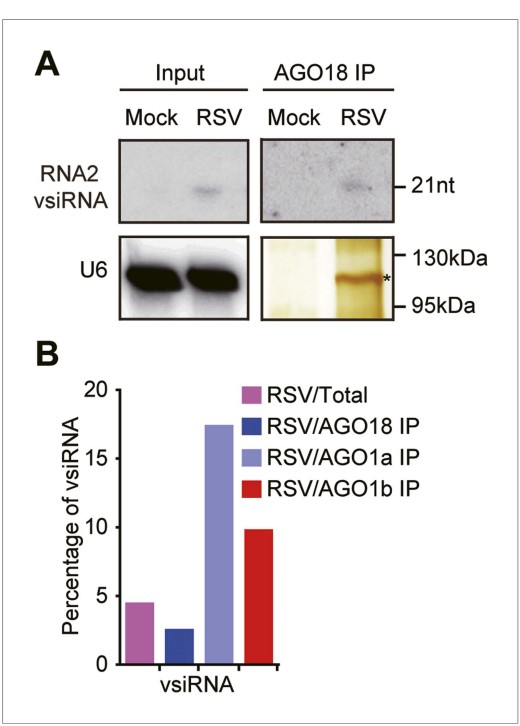

**Figure 3**. AGO18 unlikely functions as an effector of vsiRNAs. (**A**) Detection of vsiRNAs in total extracts (Input) or AGO18 immunoprecipitates prepared from mock- or RSV-inoculated plants by Northern blot. The silver-stained gel shows the quality of purified AGO18 complexes. The asterisk indicates the position of AGO18. The position of an RNA size marker is shown on the right of the blot. U6 is probed and served as a loading control. (**B**) Percentage of deep sequencing reads matching vsiRNAs in total reads obtained from total extracts, AGO1a, AGO1b, and AGO18-associated small RNAs. Samples for deep sequencing were prepared from RSV-infected rice plants.

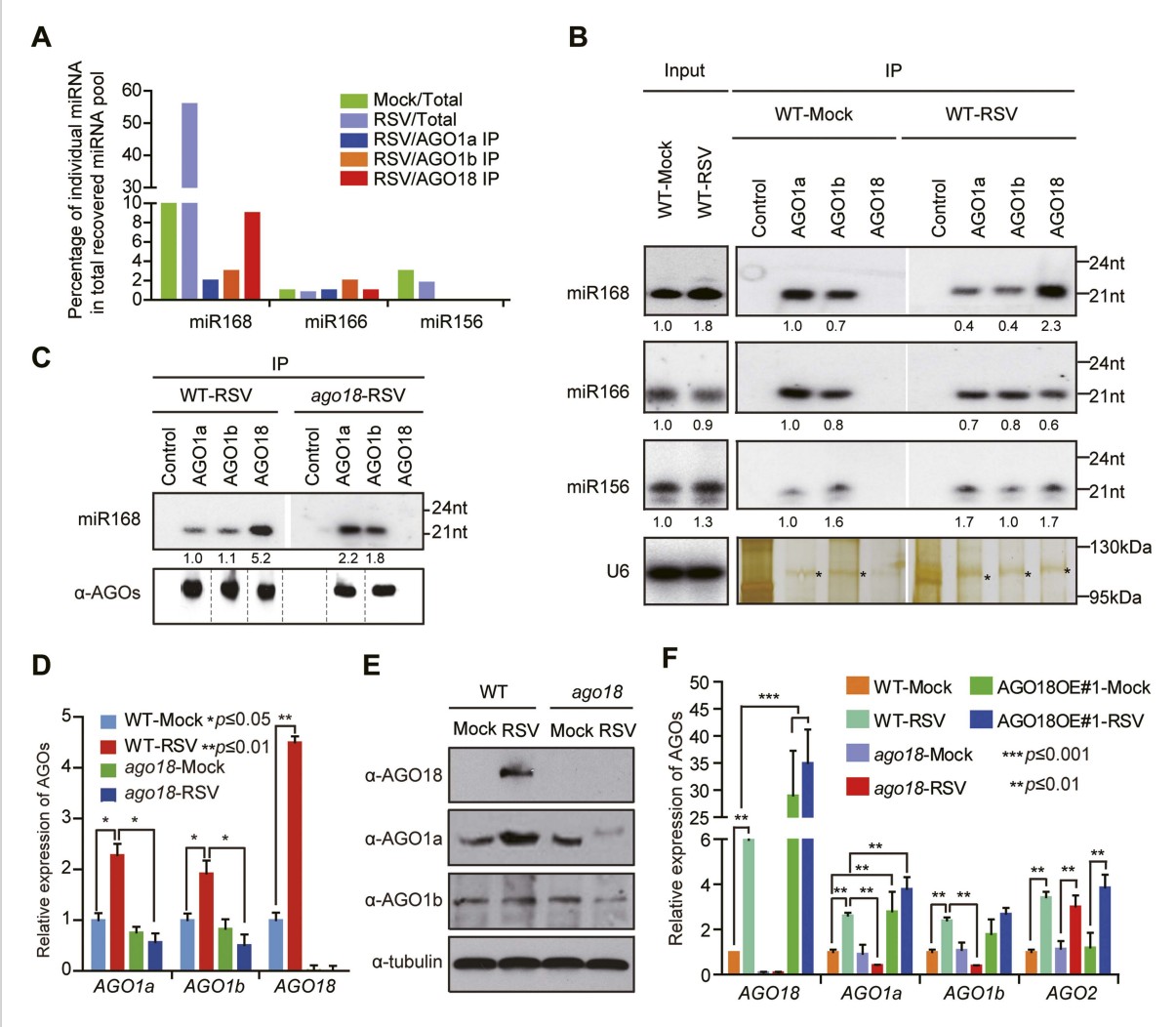

**Figure 4**. AGO18 competes with AGO1 for miR168 to up-regulate AGO1 upon viral infection. (**A**) Percentage of deep sequencing reads matching the indicated miRNAs in total reads obtained from total extracts, AGO1a, AGO1b, and AGO18-associated small RNAs. Samples for deep sequencing were prepared from mock- or RSV-inoculated rice plants. (**B**) Detection of the indicated miRNAs in total extract (Input), AGO1a, AGO1b, and AGO18 complexes by Northern blot. The blots were stripped and reprobed for multiple times. The silver-stained gel shows that comparable amounts of different AGO complexes were used for RNA preparation. The asterisks indicate the positions of AGO proteins. The positions of RNA size markers are shown on the right of the blots. The RNA signals were quantified, and the relative values were calculated by comparison with those in total extracts or AGO1a complex prepared from mock-inoculated WT (arbitrarily set to 1.0). (**C**) Northern blot analysis showing miR168 from AGO1a, AGO1b, and AGO18 complexes in RSV-infected WT and *ago18* plants (upper panel). Western blot gel shows that comparable amounts of different AGO complexes were used for RNA preparation (lower panel). (**D**) qRT-PCR analysis of the levels of *AGO1a*, *AGO1b*, and *AGO18* in WT rice and *ago18* mutants with or without RSV infection. The expression levels were normalized using the signal from *OsEF-1a*. The average (± standard deviation) values from three biological repeats of qRT-PCR are shown. (**E**) Western blot showing AGO1a, AGO1b, and AGO18 protein levels in WT and *ago18* with or without RSV infection. Tubulin was probed and served as a loading control. (**F**) qRT-PCR analysis of the levels of *AGO1a*, *AGO1b*, *AGO2*, and *AGO18* in the indicated plants. The expression levels were normalized using the signal from *OsEF-1a*. The average (± standard deviation) values from three biological repeats of qRT-PCR are shown.

To directly test whether AGO18 could indeed compete with AGO1 for binding miR168, we transiently expressed a miR168 precursor together with Flag-AGO1a or Flag-AGO1b in the presence or absence of Myc-AGO18 in *N. benthamiana* leaves. MiR444 was used as a control, because it is a species-specific miRNA in monocots that can be loaded by AGO1 (***Wu et al., 2009***), but not by AGO18 (***Supplementary file 1C***). Results from IP-northern experiments showed that when AGO18 was co-expressed with AGO1a or AGO1b, there was a nearly five fold reduction in miR168 loading

into AGO1a and AGO1b and concurrent increase in loading into AGO18 (**Figure 5**). The control miR444 was specifically loaded into AGO1a and 1b and co-expression of AGO18 did not decrease its loading into AGO1a and 1b (**Figure 5**). These data demonstrate that AGO18 can effectively compete with AGO1 for binding miR168.

## Small RNA-binding but not slicing activity of AGO18 is required for its antiviral function

The above results show that AGO18 is unlikely an effector of vsiRNAs but has the novel activity of competing with AGO1 for binding miR168. To further test this, we analyzed the functions of AGO18 domains in antiviral defense. AGO family proteins contain four characteristic domains: an N-terminal domain and conserved PAZ, MID, and PIWI domains (**Tolia and Joshua-Tor, 2007**; **Vaucheret, 2008**). The PAZ domain binds to the 3′ end of small RNAs (**Ma et al., 2004**), whereas the MID domain recognizes the 5′ end (**Kidner and Martienssen, 2005**; **Mi et al., 2008**; **Montgomery et al., 2008**; **Frank et al., 2010**). The PIWI domain adopts an RNaseH-like structure and exhibits endonuclease (slicing) activity when an Asp-Asp-His (DDH) catalytic triad is present (**Song et al., 2004**; **Rivas et al., 2005**).

Our analysis indicates that AGO18 also contains all of the four AGO signature domains and the DDH catalytic triad (**Figure 6A**). We investigated whether the small RNA binding and slicing activities

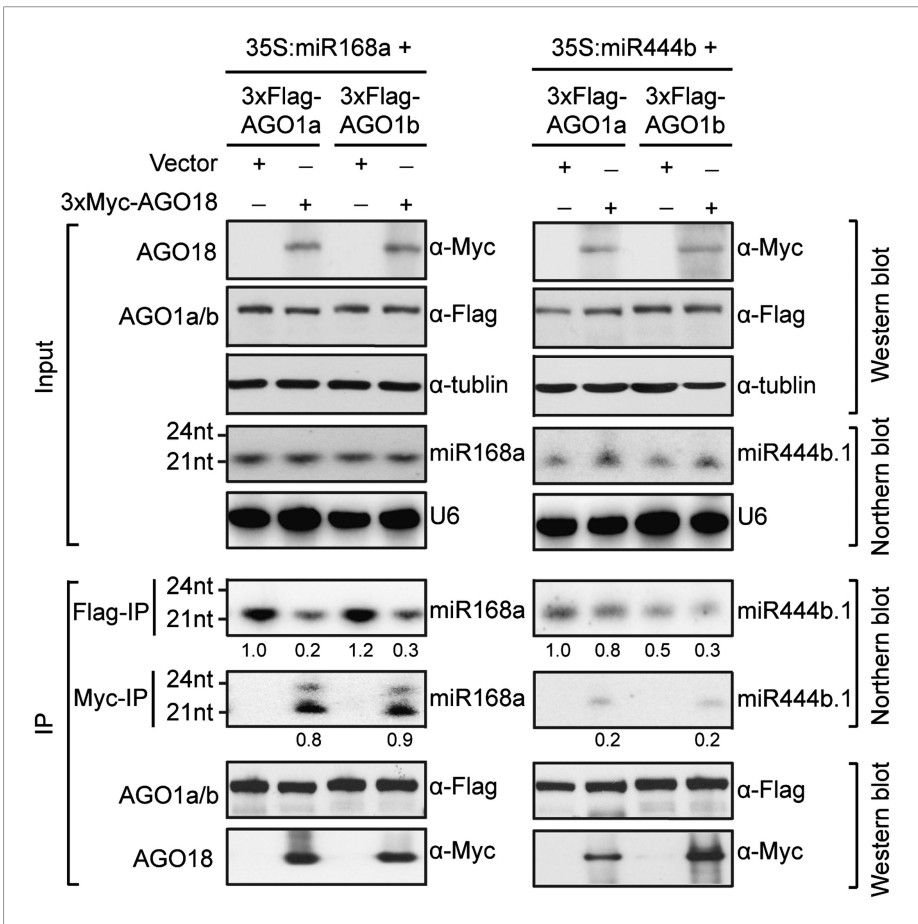

**Figure 5**. AGO18 competes with AGO1 for miR168 in vitro. Specific AGO18–miR168 interaction was confirmed by in vitro assays in *N. benthamiana*. Constructs with the indicated combinations were introduced into *N. benthamiana* leaves for transient expression by agro-infiltration. Northern blots were conducted with total RNA (Input) and small RNAs recovered from immunoprecipitated AGO complexes (IP). Western blot analyses were done with the crude extract and aliquots of the IP products using anti-Flag or anti-Myc antibodies. The positions of RNA size markers are shown on the left of the blots. U6 was probed and served as a loading control.

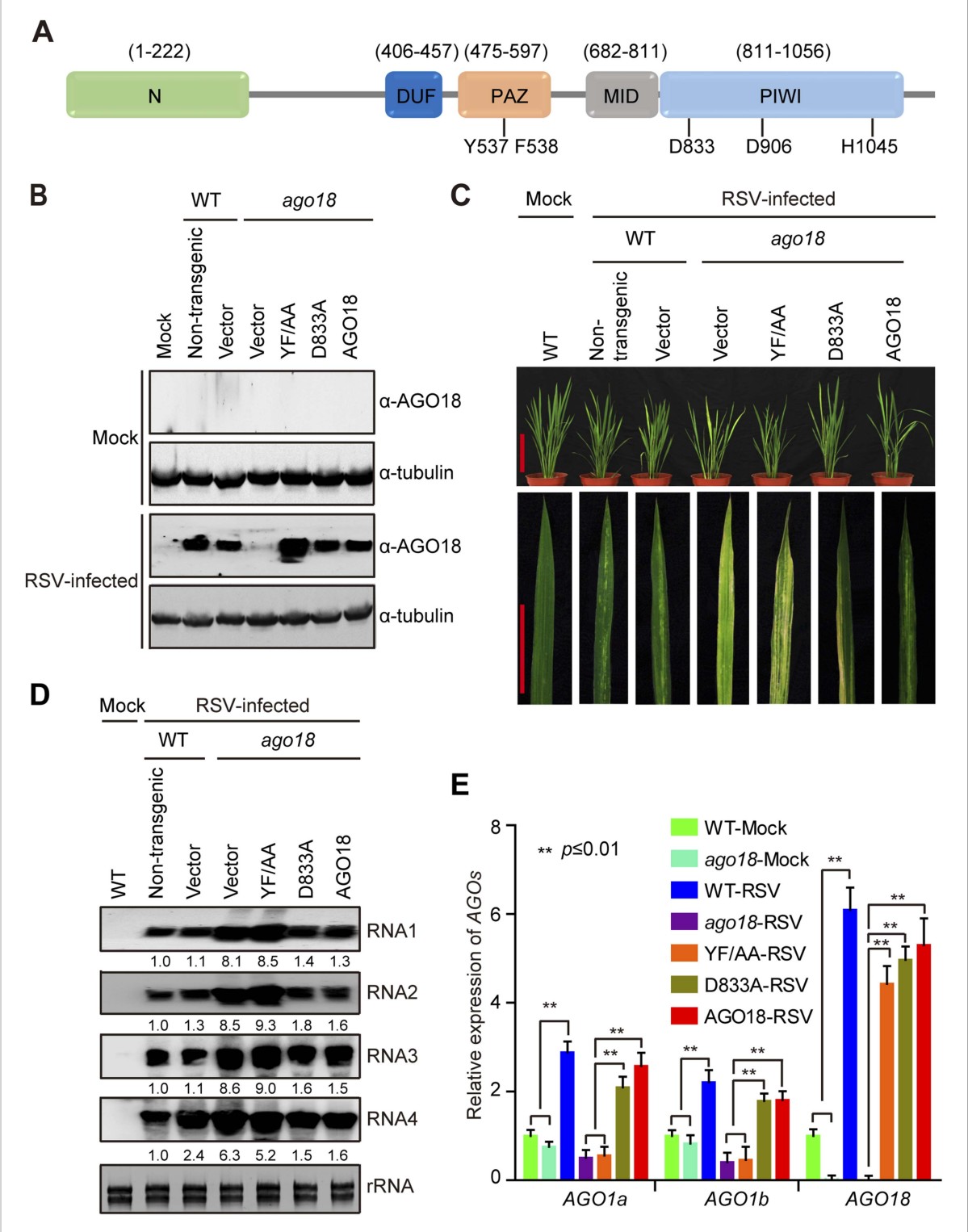

**Figure 6**. Small RNA-binding but not slicing activity of AGO18 is required for its antiviral function. (**A**) Domain structure of AGO18 protein. AGO18 consists of a variable N-terminal domain and conserved C-terminal PAZ, MID, and PIWI domains. The residues Y537 and F538 required for small RNA binding, and D833, D906, and H1045 required for slicing are indicated. (**B**) Detection of AGO18 protein in mock- or RSV-inoculated WT (Non-trangenic) plants as well as *ago18* mutants complemented with AGO18 and its derivatives by Western blot. Tubulin was probed and served as a loading control. (**C**) Symptoms of mock- or RSV-inoculated WT (Non-trangenic) plants as well as *ago18* mutants complemented with AGO18 or its derivatives, pictures were taken at 6 weeks post-inoculation. Scale bars, 15 cm (upper panel) and 5 cm (lower panel). (**D**) Detection of RSV genomic RNA segments in mock- or RSV-inoculated

*Figure 6. continued on next page*

*Figure 6. Continued*

WT (Non-trangenic) plants as well as *ago18* mutants complemented with AGO18 or its derivatives by Northern blot. The blots were hybridized with radiolabeled riboprobes specific for each RNA segment. rRNAs were stained with ethidium bromide and served as loading controls. The RNA signals were quantified and normalized to rRNAs, and the relative values were calculated by comparison with those in RSV-infected WT (arbitrarily set to 1.0). (**E**) qRT-PCR analysis of the levels of *AGO1a*, *AGO1b*, and *AGO18* in RSV-infected WT plants as well as *ago18* mutants complemented with AGO18 or its derivatives. The expression levels were normalized using the signal from *OsEF-1a*. The average (± standard deviation) values from three biological repeats of qRT-PCR are shown.

of AGO18 are required for its role in antiviral defense by transgenic complementation experiments. We generated two AGO18 mutants, AGO18$^{Y537A/F538A}$ (YF/AA) and AGO18$^{D833A}$ (D833A). YF/AA contains alanine substitutions at two conserved residues (Y537F538) that are known to be required for small RNA binding (*Ma et al., 2004*; *Guang et al., 2008*; *Ye et al., 2012*), whereas D833A contains an alanine substitution at the DDH catalytic triad (*Song et al., 2004*; *Rivas et al., 2005*; *Wee et al., 2012*). We transgenically expressed WT AGO18, YF/AA, and D833A mutants under the control of the native *AGO18* promoter in the *ago18* mutant background. We then inoculated these plants with RSV by insect vector transmission. Western blots confirmed the expression of WT and mutant AGO18 in the infected transgenic plants (*Figure 6B*). Based on the disease symptoms (*Figure 6C*) and accumulation of viral genomic RNAs (*Figure 6D*), both WT AGO18 and D833A could mostly complement the *ago18* mutant for resistance to RSV infection, whereas the YF/AA mutant could not. These results suggest that AGO18 exerts its antiviral function mainly through small RNA binding, rather than slicing. We also measured the expression levels of *AGO1* in these transgenic lines that were infected by RSV. We found that *AGO1* expression was elevated by viral infection in the transgenic plants that express WT AGO18 or D833A mutant but not in those that express YF/AA (*Figure 6E*), further suggesting that the small RNA binding but not slicing activity of AGO18 is required for its role in up-regulating AGO1.

## Expression of miR168-resistant *AGO1a* rescues the deficiency of *ago18* for viral resistance

The above studies demonstrated that AGO18 regulates AGO1 homeostasis by sequestering miR168 during viral infection. To further test this, we generated transgenic rice plants expressing wild type (AGO1a lines) and miR168-resistant *AGO1a* (AGO1a-Res lines) in the *ago18* background, under the control of the native *AGO1a* promoter (*Figure 7A*). The steady-state *AGO1a/AGO1a-Res* mRNA levels of both AGO1a and AGO1a-Res transgenes accumulated to higher levels relative to the *AGO1a* levels in wild-type (WT) non-transgenic and *ago18* plants (*Figure 7B*). It is noteworthy that *AGO1a* mRNA level was reduced by RSV infection in the AGO1a transgenic line, whereas AGO1a-Res transgenic lines had no significant reduction in *AGO1a-Res* mRNA levels upon RSV infection (*Figure 7B*), indicating that AGO1a is subject to regulation by miR168/AGO18. Although the transcript levels of the miR168-resistant transgene *AGO1a* markedly increased, none of these lines showed notable phenotypic differences from WT or *ago18* plants in growth and development. These transgenic lines, however, were much more resistant to RSV infection than *ago18* and AGO1a plants (*Figure 7B,C*). Viral replication significantly decreased in the AGO1a-Res plants (*Figure 7D*). Thus, AGO1 over-expression could rescue the deficiency of *ago18* for viral resistance. These data provided further compelling evidence that AGO18 functions through sequestering miR168 to up-regulate AGO1 against virus infections.

Based on the above results, we conclude that AGO18 regulates AGO1 homeostasis by specifically loading miR168 during viral infection. Sequestering of miR168 by AGO18 leads to increased accumulation of AGO1 to embark on an effective antiviral defense response.

## Discussion

AGO proteins are at the heart of RNAi machinery. Building upon the conserved core components of the RNA silencing machinery such as DCLs and AGOs, plants have evolved extensive variants of these components, especially AGOs. This evolutionary amplification of the machinery components implies amplification/diversification in regulatory mechanisms/functions. While the roles of some AGOs in

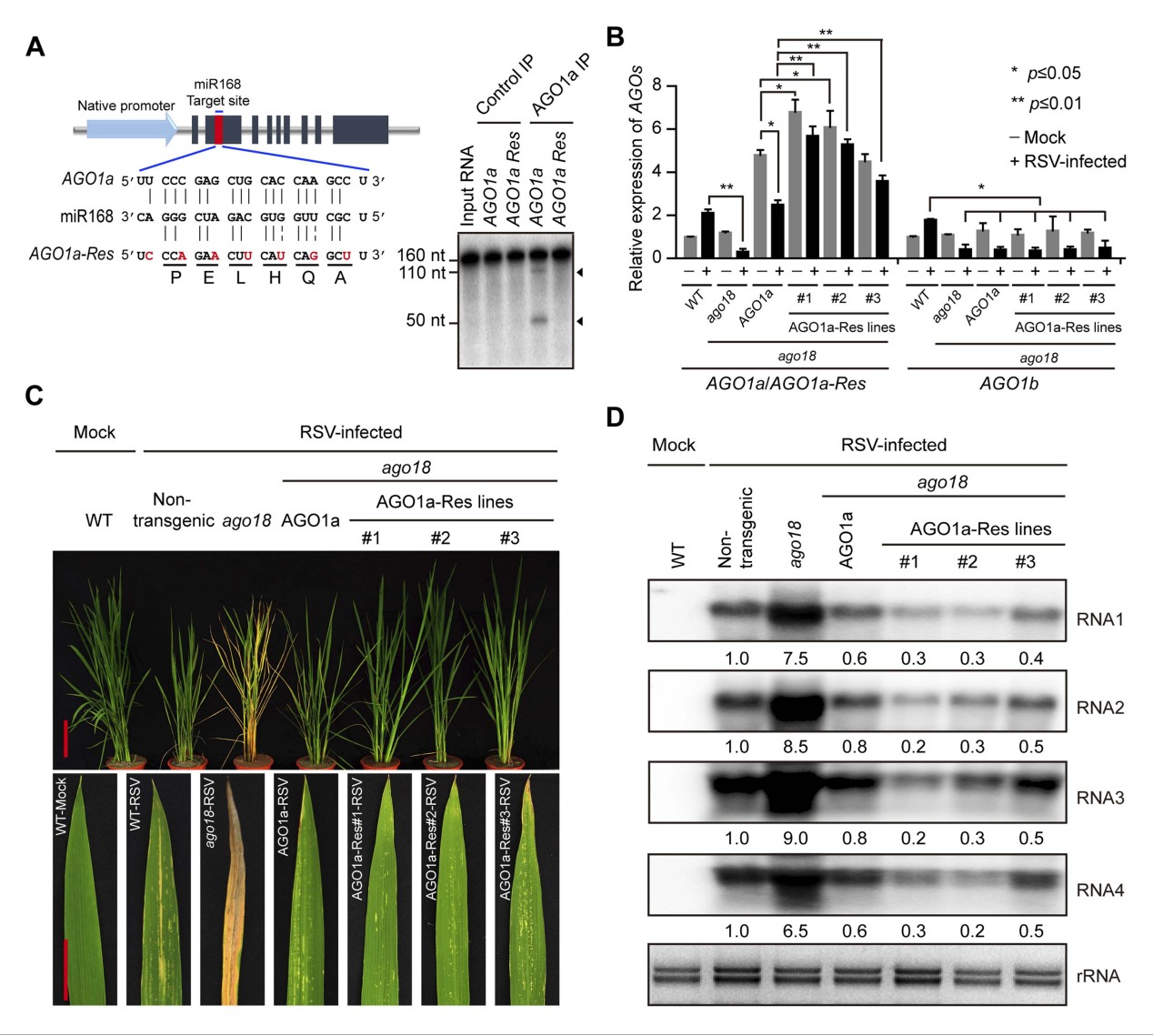

**Figure 7**. Transgenic expression of miR168-resistant AGO1a rescued the deficiency of *ago18* for viral resistance. (**A**) Schematic drawing of AGO1a featuring the target site of miR168. In *AGO1a* that is resistant to miR168 cleavage (*AGO1a-Res*), seven synonymous nucleotide substitutions were introduced into the miR168 target site. The resistance of *AGO1a-Res* to miR168-directed cleavage was examined by in vitro cleavage assay using purified AGO1a complex. Positions of the cleavage products are indicated by arrows. (**B**) qRT-PCR analysis of *AGO1a* and *AGO1b* expression levels in mock (−) or RSV-inoculated (+) WT, *ago18*, and transgenic plants expressing *AGO1a* or *AGO1a-Res* in the *ago18* mutant background. The expression levels were normalized using the signal from *OsEF-1a*. The average (± standard deviation) values from three biological repeats of qRT-PCR are shown. (**C**) Symptoms of mock- or RSV-inoculated WT (Non-transgenic), *ago18*, and transgenic plants expressing *AGO1a* or *AGO1a-Res* in the *ago18* mutant background. Scale bars, 15 cm. (**D**) Detection of RSV genomic RNA segments in the indicated plants by Northern blot. The blots were hybridized with radiolabeled riboprobes specific for each RNA segment. rRNAs were stained with ethidium bromide and served as loading controls. The RNA signals were quantified and normalized to rRNAs, and the relative values were calculated by comparison with those in RSV-infected WT (Non-trangenic) (arbitrarily set to 1.0).

specific functions of RNA silencing in gene regulation and innate immunity have been well studied, the roles of most AGOs remain unknown or poorly understood. Besides AGO1, only the slicing function of AGO2 appears to be also important for antiviral defense in *Arabidopsis* (*Harvey et al., 2011*; *Jaubert et al., 2011*; *Scholthof et al., 2011*; *Wang et al., 2011*; *Carbonell et al., 2012*; *Xia et al., 2014*). Current studies on the roles of several *Arabidopsis* AGOs in antiviral defense are dictated by the target RNA-slicing paradigm.

In the arms race between host defense and viral counter-defense, plant viruses have evolved various strategies, including encoding suppressors of RNA silencing, to interfere with different steps of the host RNA silencing defense pathways (*Ding and Voinnet, 2007*; *Ding, 2010*). In *Arabidopsis* and *N. benthamiana*, infection by many viruses leads to increased levels of *AGO1* mRNA readying the plants to combat viruses, but the viruses also simultaneously increase the expression levels of miR168 to down-regulate AGO1 to defeat host defense (*Várallyay et al., 2010*). RSV infection of rice also leads to an increase in miR168 (*Figure 4A,B*) (*Du et al., 2011*). Thus, elevated miR168 expression appears to be a broad mechanism of viral counter-defense.

In this study, we discovered a novel mechanism of positive regulation of AGO1 activity by AGO18 in antiviral RNAi. We showed that AGO18 does not have antiviral function by itself, but its presence is required for the antiviral function of AGO1. AGO18 accomplishes this not via its slicing function but through its competition with AGO1 for binding miR168. This binding inhibits miR168-mediated auto-regulation of AGO1, thereby boosting the accumulation levels of AGO1 to enable antiviral function. Thus, AGO18 has evolved as a novel and dedicated positive regulator of the plant surveillance/defense system. Our findings suggest that AGO18-bound miR168 is not competent in cleaving *AGO1* mRNA in RSV-infected rice plants, albeit AGO18 contains conserved DDH motif for slicing activity. Several possible mechanisms can be considered. First, the catalytic activity of AGO18 may not be as potent as that of AGO1 due to some sequence/structure differences between these two AGOs. Second, the slicing activity of AGO18 may be repressed by a yet-to-be-identified protein that specifically interacts with AGO18. Third, the cellular compartmentation of AGO18/miR168 complex may be distinct from that of AGO1/miR168 and prevent its access to *AGO1* mRNA. Finally, AGO18 binding of miR168 may subsequently induce the degradation of miR168. It will be highly interesting to test these possibilities in future studies.

There were previous examples of competitive binding of other miRNAs between other AGOs and AGO1 in *Arabidopsis*. miR168 can be incorporated into AGO10 to decrease the translation efficiency of *AGO1* mRNA in *Arabidopsis* (*Mallory et al., 2009*). The *Arabidopsis* miR166/165 are significantly enriched in AGO10-bound miRNAs, preventing them from being loaded into AGO1 to fulfill their normal roles in development (*Zhu et al., 2011*; *Ji et al., 2011*; *Manavella et al., 2011*). MiR390 is associated with AGO7 to mediated trans-acting siRNA biogenesis (*Montgomery et al., 2008*), through cooperative activity of AGO1. In addition, miR408 associates with both AGO1 and AGO2 redundantly to regulate *Plantacyanin* mRNA levels (*Maunoury and Vaucheret, 2011*). This competition, however, does not affect AGO1 homeostasis. In contrast to these negative regulations of AGO1, binding of miR168 by AGO18 in infected rice plants boosts AGO1 accumulation. Thus, among mechanisms of AGO-regulation of AGO1 homeostasis or activity, the up-regulation of AGO1 via AGO18 binding of miR168 represents a novel type of mechanism. Whether other AGOs function similarly in *Arabidopsis*, rice, and other plants to up-regulate AGO1 or another AGO to impact developmental processes or defense responses is an outstanding question to be addressed in future studies. It is also noteworthy that, in addition to miR168, several miRNAs including miR528, miR159a, and miR159b were also recruited by AGO18 in RSV-infected rice plants (*Supplementary file 1C*). Thus, for AGO18, we cannot rule out the possibility that AGO18 plays regulatory roles in other capacities or has its own independent biological functions.

It is important to emphasize that most plant viruses are transmitted to plants by insect vectors under natural infection conditions in the field, and yet the vast majority of studies so far on the mechanisms of RNA silencing-mediated antiviral responses employed infection methods such as mechanical inoculation with in vitro viral RNA transcripts or virions and infiltration with agrobacteria carrying engineered viral DNAs. These studies often used viral delivery methods with much higher levels of inoculum than natural field conditions. How some of such manipulations would alter host responses remains an outstanding question. Here, we used insect vectors that carry the natural forms of viruses to inoculate plants, best reflecting what happens under field infection conditions with regard to the behavior of viruses and host responses. This is not only important to dissect the natural infection and defense mechanisms, but also important for developing effective technologies to combat viral infection under natural infection conditions. Recent studies indicated that natural infection in early stage of virus infection may trigger different host-defense reaction (*Garcia et al., 2014*).

AGO18 is of special interest because it has evolved as a conserved clade in monocots that encompass many important crop plants for foods and biofuels. Whether it plays a similar role in

antiviral defense in other monocots warrants further investigations. From a broader perspective, we expect that further studies on the extensive RNA silencing machinery components evolved in crop plants may lead to new discoveries about their functions in shaping plant diversity with regard to phenotypes and innate immunity mechanisms. Finally, sequestering miR168 by AGO18 to regulate AGO1 homeostasis represents an evolutionary novelty. It raises the question of whether small RNA competition-based regulations between other types of proteins have also evolved to regulate different biological processes in plants and other organisms. This mechanism functions against infection by two evolutionarily distinct viruses, and likely has broader significance in resistance against more viruses in a wide range of monocots. We propose that engineering rice and other cereals to over-express AGO18 may provide a new strategy for the control of diverse viral pathogens.

## Materials and methods

### Plant growth and virus inoculation

Plant growth and virus inoculation were essentially carried out as described (*Du et al., 2011*). Briefly, rice (*O. sativa* spp. *japonica*) seedlings were grown in a greenhouse at 28–30°C and 60 ± 5% relative humidity under natural sunlight for 4 weeks. The viruliferous (RSV and RDV-carrying) insects of *L. striatellus* and *N. cincticeps*, as well as virus-free *L. striatellus* and *N. cincticeps* (mock) were used for inoculation. After feeding 3 days, the insects were removed, and the rice seedlings were returned to the greenhouse to grow under the greenhouse conditions above. 3 weeks post-inoculation when the newly developed leaves started to exhibit viral symptoms, the whole seedlings were harvested. For each sample, at least 15–20 rice seedlings were pooled for RNA extraction.

### Non-preference test

Non-preference tests were performed for all rice seedlings of the different genetic backgrounds with the two viral transmission insect vectors brown planthopper (*L. striatellus*) and leafhopper (*N. cincticeps*). Details of the procedures were described previously (*Hiroshi et al., 1994*).

### Histochemical GUS staining

Plants were infiltrated with 50 mM sodium phosphate (pH 7.0), 10 mM EDTA, and 0.5 mg/ml X-gluc (Apollo Scientific, UK), followed by incubation at 37°C in the dark overnight and then destained in 70% ethanol before photographing.

### Generation of antibodies against rice AGOs

Synthetic peptides AGO1aN (KKKTEPRNAGEC), AGO1bN (KKRTGSGSTGEC), and AGO18N (YHGDGERGYGRC) were used to raise rabbit polyclonal antibodies against AGO1a, AGO1b, and AGO18, respectively, essentially as described (*Wu et al., 2009*; *Mi et al., 2008*). The antisera were affinity purified and used for immunoprecipitation (IP) (1:50 dilution).

### Constructs and transgenic lines

Gateway system (Invitrogen, Carlsbad, CA) was used to make binary constructs. Several destination vectors were created for transient expression in *N. benthamiana* and stable rice transformation. The binary gateway vector pMDC32 (*Karimi et al., 2007*) was modified to obtain rice AGO18 promoter (*p32:pAGO18*) or AGO1a promoter (*p32:pAGO1a*). Most cDNA and miRNA genes were cloned into pENTR/D vectors and *pENTR/D-Flag-AGO18D833A, YF/AA and AGO1a:miR168 Res* clones were prepared by using the QuikChange site-directed mutagenesis kit (Stratagene, La Jolla, CA). All these clones were confirmed by sequencing and transferred to the appropriate destination vectors by recombination using the Gateway LR Clonase II Enzyme mix (Invitrogen). *pCam2300:Actin1::OCS* and *pCam2300:35S::OCS* vectors (*Wu et al., 2009*) were used to generate *pCam2300:Actin1::Flag-AGO18*, *pCam2300:35S::Myc-AGO18*, *Flag-AGO1a*, *Flag-AGO1b*, *miR168a*, and *miR444b*, respectively. All PCR primers are listed in *Supplementary file 1D*.

### Transient expression in the *N. benthamiana*

Assays of transient expression in leaves of *N. benthamiana* were performed as described (*Voinnet et al., 2003*). A detailed protocol is available upon request.

## Western blotting

Protein samples were boiled with same volume of 2× protein loading buffer at 95°C for 5 min and separated by SDS-PAGE gel. Proteins were then transferred to PVDF membranes and detected with antibodies against AGO18, AGO1a, AGO1b, MYC (11667203001, Roche, Switzerland), FLAG (F1804, Sigma–Aldrich, St. Louis, MO), and tubulin (T5168, Sigma–Aldrich).

## Purification of rice AGO-containing complexes and associated small RNAs

Rice AGO-containing complexes were immunopurified from RSV-inoculated rice plants as previously described (*Qi et al., 2005*; *Wu et al., 2010*). The quality of purification was examined by SDS-PAGE followed by silver staining or IP-western blotting, and the bands of expected sizes were confirmed as AGO proteins by mass spectrometry. RNAs were isolated from total cell extracts and from the purified AGO complexes by Trizol reagent (Invitrogen), resolved on a 15% denaturing PAGE gel, and visualized by SYBR-gold (Invitrogen) staining. Gel slices within the range of 18–28 nucleotides were excised, and the RNAs were eluted and purified for cloning.

## Small RNA cloning and sequencing

Small RNA cloning for Illumina sequencing was carried out essentially as described previously (*Mi et al., 2008*; *Wu et al., 2010*). A detailed protocol is available upon request.

## Small RNA Northern blotting

Northern blot analysis with total sRNAs or from purified AGO complexes was performed as described before (*Qi et al., 2005*). $^{32}$P-end labeled oligonucleotide probes complementary to sRNAs were used for Northern blots. The sequences of the probes are listed in *Supplementary file 1D*.

## Quantitative RT-PCR analysis

Total RNAs were extracted from rice plants with Trizol (Invitrogen). After removal of contaminating DNAs by digestion with RNase-free DNaseI (Promega, Madison, Wisconsin, USA), the RNAs were reverse transcribed by SuperScript III Reverse Transcriptase (Invitrogen) using oligo (dT). The cDNAs were then used as templates for quantitative PCR and RT-PCR. Quantitative PCR was performed using SYBR Green Real-time PCR Master Mix (Toyobo, Osaka, Japan). The rice *OsEF-1a* gene was detected in parallel and used as the internal control. All the other primers used are listed in *Supplementary file 1D*.

## Bioinformatics analysis of small RNA data sets

The adaptor sequences in Illumina 1G sequencing reads were removed by using 'vectorstrip' in the EMBOSS package. The sRNA reads with length of 19–27 nt were mapped to the rice nuclear, chloroplastic, and mitochondrial genomes (http://rice.plantbiology.msu.edu/, version 6.0). The sRNAs with perfect genomic matches were used for further analysis. Rice miRNA annotations were from miRBase (http://microrna.sanger.ac.uk/sequences, Release14) and our previous publication (*Wu et al., 2009*; *Du et al., 2011*). Statistical analysis of the sRNA data sets was done by using in-house-developed Perl scripts (*Supplementary file 2*).

## miRNA cleavage assay

miRNA cleavage activity assay was performed essentially as described (*Qi et al., 2005*; *Qi and Mi, 2010*), using immunopurified rice AGO1a complex and in vitro-transcribed *AGO1a* or *AGO1a-Res* transcripts.

## Accession numbers

Small RNA data sets generated in this study are deposited in the NCBI sequence read archive (SRA) (http://www.ncbi.nlm.nih.gov/sra) under accession number PRJNA273330.

## Acknowledgements

We thank the National Institute of Biological Sciences (NIBS) Mass Spectrometry Center, Beijing, for sequencing AGO18 protein, NIBS Antibody Center and Integrated R&D Services—WuXi AppTec for generating the antisera used in this study. This work was supported by grants from the National Basic

Research Program 973 (2014CB138400, 2011CB100703 and 2013CBA01403), Natural Science Foundation of China (31225015, 31421001, 31030005, 31420103904, and 31272018), Transgenic Research Program (2014ZX08010-001), and Doctoral Fund of Ministry of Education of China (20113515120004). JGW was supported in part by the Postdoctoral Fellowship of Peking-Tsinghua Center for Life Sciences.

## Additional information

### Funding

| Funder | Grant reference | Author |
|---|---|---|
| Ministry of Science and Technology of the People's Republic of China | National Basic Research Program 973, 2014CB138400 | Yijun Qi, Yi Li |
| National Natural Science Foundation of China | 31030005 | Jianguo Wu, Yi Li |
| National Natural Science Foundation of China | 31420103904 | Jianguo Wu, Yi Li |
| Ministry of Science and Technology of the People's Republic of China | National Basic Research Program 973, 2011CB100703 | Yijun Qi |
| Ministry of Science and Technology of the People's Republic of China | National Basic Research Program 973, 2013CBA01403 | Jianguo Wu, Yi Li |
| National Natural Science Foundation of China | 31421001 | Yijun Qi |
| National Natural Science Foundation of China | 31225015 | Yijun Qi |
| National Natural Science Foundation of China | 31272018 | Jianguo Wu |
| Transgenic Research Program, Ministry of Agriculture of the People's Republic of China | 2013ZX08010-001 | Yi Li |

The funders had no role in study design, data collection and interpretation, or the decision to submit the work for publication.

### Author contributions

JW, ZY, Conception and design, Acquisition of data, Analysis and interpretation of data, Drafting or revising the article; YW, LZ, YJ, SZ, SJ, RL, LX, HZ, Acquisition of data, Analysis and interpretation of data; RY, Acquisition of data, Analysis and interpretation of data, Contributed unpublished essential data or reagents; YZ, XZ, Acquisition of data, Contributed unpublished essential data or reagents; XC, Analysis and interpretation of data, Contributed unpublished essential data or reagents; LX, Analysis and interpretation of data, Drafting or revising the article; ZW, Conception and design, Analysis and interpretation of data, Drafting or revising the article; YL, YQ, Conception and design, Analysis and interpretation of data, Drafting or revising the article, Contributed unpublished essential data or reagents

## Additional files

### Supplementary files

• Supplementary file 1. (**A**) Disease incidence of RSV- or RDV-inoculated rice plants. (**B**) Summary of small RNA deep sequencing datasets. (**C**) Analysis of miRNA abundance by deep sequencing. (**D**) Oligonucleotides used in this study.

• Supplementary file 2. Perl script used for statistical analysis of miRNA expression.

## Major dataset

The following dataset was generated:

| Author(s) | Year | Dataset title | Dataset ID and/or URL | Database, license, and accessibility information |
|---|---|---|---|---|
| Wu J, Yang Z, Wang Y, Zheng L, Ye R, Ji Y, Zhao S, Ji S, Liu R, Xu L, Zheng H, Zhou Y, Zhang X, Cao X, Xie L, Wu Z, Qi Y and Li Y | 2015 | Small RNA populations in Argonaute complexes purified from Rice stripe Tenuivirus (RSV)-infected rice | http://www.ncbi.nlm.nih.gov/bioproject/273330 | Publicly available at NCBI Sequence Read Archive (PRJNA273330). |

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
