## [Decision Letter]

[Editors’ note: a previous version of this study was rejected after peer review, but the authors submitted for further consideration. The first decision letter after peer review is shown below.]

Thank you for choosing to send your work entitled “Antiviral RNAi requires a novel cooperative activity of two distinct Argonaute proteins” for consideration at *eLife*. Your full submission has been evaluated by Detlef Weigel (Senior editor) and three peer reviewers, one of whom is a member of our Board of Reviewing Editors, and the decision was reached after discussions between the reviewers. We regret to inform you that your work will not be considered further for publication.

All three reviewers have major but non-overlapping concerns about the manuscript. The major reservations are that:

1) The concept described overlaps with the previously described AGO1-AGO10 competition;

2) The study is incomplete. Some obvious experiments have not been done include (but are not restricted to) the use of miR168 resistant AGO1 mRNA and analysis of AGO2;

3) The fact that all RNAi lines (DCL1, DCL2, DCL3a, DCL3b, DCL4, AGO1) have the same phenotype, raising the possibility that there is interference of transgene RNAi and antiviral RNAi.

*Reviewer #1*:

The paper confirms that RNA silencing mediates defense against RSV and RDV delivered by insect vectors. Figure 1 confirms that several DCL proteins and AGO1 are required as in the canonical RNA silencing pathway. Figure 2 data implicates a monocot conserved AGO18 in this antiviral silencing. The AGO18 promoter is induced in infected plants and the mutants are hypersusceptible to virus infection. Conversely the overexpressing plants are less susceptible to infection than the control. However the AGO18 does not bind viral siRNA as effectively as AGO1 and a simple role of this unusual AGO in straightforward RNA silencing is ruled out. The AGO18 bound RNAs are instead enriched for miR168 relative to AGO1. The role of miR168 is to silence the mRNA for AGO1 and so the data are consistent with a model in which AGO18 is a negative regulator of a negative regulator of AGO1. Data to support this model are from: Competition experiments showing that overexpression of AGO18 competed with AGO1 for binding of overexpressed miR168 but not miR443 and from the findings that a mutant AGO18 with defective miRNA binding was not able to complement the mutant phenotype but the protein with inactivating mutations in the nuclease active site did.

Overall these are nicely executed experiments although I would say that the obvious and most direct test of the hypothesis with a miR168 resistant AGO1 message is conspicuously absent. The data also do not test the prediction that AGO18 induction in virus-infected plants would mediate overexpression of AGO1 and that the timing of the effect would precede the maximal rate of virus accumulation. The finding that the plant AGO-mediated antiviral silencing can be enhanced in virus-infected plants is not new and the decoy AGO mechanism is also not novel although it was described previously in a different context. Taking the sum of these considerations my view is that the paper in its present form is more suited to a specialist journal.

There are several interesting features in this paper that could be followed up to make a new paper that might be more appropriate to *eLife*. It would be interesting to know for example about the features of AGO18 that differentiate it from other AGO clades and to know more about the regulation of AGO18 in virus-infected plants. Specific questions would concern the role of insect vectors in the activation of this gene and the possibility (implied by Figure 2) that there is a systemic signal associated with its induction. We know about signals induced in disease resistant plants but this would be a signal in susceptible plants. Is it an RNA or perhaps it is a JA response induced by the insect vectors of the virus?

*Reviewer #2*:

In this manuscript, Wu et al, show that AGO18, a monocot-specific member of the ARGONAUTE family, is induced during virus infection and preferentially binds to miR168, a miRNA that normally down-regulates AGO1, which has antiviral RNAi activity. Thus, AGO18 induction titrates miR168, leading to up-regulation of AGO1 and increased resistance to viruses. The data are convincing and support the authors' conclusions.

However, the authors totally ignore the contribution of AGO2, both in the bibliographic introduction and in the development of their story. Indeed, at least five papers have shown that, in addition to AGO1, AGO2 contributes to antiviral RNAi activity (16; 51; 43; 20; 4). None of these papers is referenced in the Introduction and only a brief sentence in the Discussion says that, in addition to AGO1, AGO2 also has catalytic activity against viral RNA, but there is no reference cited. The absence of investigation on AGO2 in this paper is a big concern, considering that the authors have previously shown that AGO2 is induced during virus infection, and actually more induced than AGO1 (11). Could AGO18 induction also lead to AGO2 up-regulation, and by which mechanism? This question needs to be addressed.

In addition, the results obtained with the *dcl* RNAi lines are puzzling and actually raise a number of concerns regarding the RNAi strategy. Indeed, RNAi against either DCL1, DCL2, DCL3a, DCL3b or DCL4 increases infection rates. How is this possible, given the different role of each DCL? Isn't it that the production of very large amounts of dsRNA during RNAi somehow titrates the DCL(s) required for antiviral activity? If so, the results obtained during AGO1 RNAi are also questionable.

*Reviewer #3*:

Small RNAs (sRNAs) including miRNAs and siRNAs are a group of non-coding RNAs that regulate expression of complementary target genes. sRNAs are loaded into AGO-centered RNA inducing silencing complexes (RISCs) and guide AGO to cleave their targets. It is well-known that plants, and possibly other organisms as well, utilize RNAi system to defense themselves from virus propagation. On the other hand, viruses encode viral suppressors to block different steps of posttranscriptional gene silencing pathway (i.e. destabilizing and inhibiting AGO proteins) to counter defense the host responses. It has been also shown that some viruses evolve a mechanism that induces expression of endogenous miR168 that happens to target AGO1 effector itself, leading to suppression of RNAi so that viruses can survive.

In this paper, Wu and collaborators studied siRNA responses of plants to the RSV infection, the authors took advantage of the RSV/RDV–Oryza sativa patho-systems. These RNA viruses infect rice by a planthopper vector and establish infection by overcoming the AGO1-mediated antiviral siRNA response in the plant. Nonetheless, the sRNA machinery is highly diversified in rice, encoding 5 DCL and 18 AGO proteins, most of which have still unknown functions. Importantly, the authors opted to use the natural insect vector of the viruses in the inoculation of the plants to mimic the natural infection conditions and be able to discern important effectors in the sRNA mediated antiviral defense. They found that one AGO protein, AGO18, is specifically upregulated by virus infection and also essential for antiviral immunity. They then tried to explore the mechanism underlying AGO18 function. They found that AGO18 bound less viral-derived siRNAs compared to canonical AGO1, leading to their proposal that AGO18 is unlikely an effector protein for viruses. Because they previously published a paper that the viral infection elevate the expression of both miR168 and AGO18, they hypothesized there might be come connection between the two events. They performed a deep-seq and found that AGO18 bound a large amount of miR168 (but not the largest group), and therefore proposed a model and evidence that Ago18 maintains AGO1 level by sequestering miR168 as an antiviral strategy.

Overall, the manuscript is interesting and presented lots of nice data, most of which are likely solid. It will advance the field, especially, understanding of the arms race between hosts and pathogens. This manuscript is amazingly an extension of a previous story that AGO10 specifically sequesters miR165/166 to antagonize the silencing activity of AGO1 ([59]. Cell). However, there are some issues in the manuscript, some of them might be serious if not solved.

1) The authors should integrate their work in the literature context. Starting from the Introduction of well-studied AGO functions, the authors selectively emphasized their own work and ignored other elegant work (i.e. AGO7-miR390; Montgomery, et al.; 2008). Another example is that the mechanism presented here is essentially the same as the AGO10-AGO1 story (59). The only difference is AGO18-AGO1 for miR168 (which happens to feedback regulate AGO1). The way the authors wrote the manuscript suggests that their discovery is completely novel, if one does not go deep through the Discussion part.

2) The manuscript missed one piece of important information: could sequestering of miR168 under AGO18 promoter through miRNA sponge or target mimicry rescue ago18 responses to viral infection?

3) In Figure 2 it is apparent an increase in AGO18 levels upon RSV infection, especially in line #5. This is unexpected given that the gene is under a constitutive promoter. Is it possible that the virus infection affects the protein stability? It is known that AGO proteins are unstable in the apo form (this is without sRNA bound), could it be possible that virus miR168 induction regulates AGO18 stability? The authors could address this issue by checking transcript and protein levels of AGO18 in OE lines before and after virus infection. Is the concurrent increase of AGO18 and miR168 upon viral infection synergetic effect?

4) In the [Supplementary-material SD1-data], the highest accumulation of miRNA in AGO18 complexes is of miR528, which is more than 9-fold higher than in *AGO1b* or ∼3-fold *AGO1a* under virus infection. Given that miR528 is an important metabolic regulator during embryo development in rice, it seems very interesting to discuss its possible relationship to virus defense and whether AGO18 acts also as a decoy and not a silencing effector with this miRNA.

5) In the Discussion part, the authors attempted to differentiate this finding from the AGO10-AGO1 weakens, instead of strengthening, their discussion, for having evidence of this phenomenon used not only for development but as an adaptive mechanism for viral defense in a distant organism as rice, makes it ubiquitous and conserved.

[Editors' note: further revisions were requested prior to acceptance, as described below.]

Thank you for resubmitting your work entitled “Viral-Inducible Argonaute18 Confers Broad-Spectrum Virus Resistance in Rice by Sequestering A Host MicroRNA” for further consideration at *eLife*. Your revised article has been favorably evaluated by Detlef Weigel (Senior editor) and three reviewers, one of whom, David Baulcombe, is a member of the Board of Reviewing Editors. The manuscript has been improved but there are some remaining minor issues that need to be addressed before final acceptance, as outlined below:

1) We strongly recommended removal of the *dcl* mutant analysis as the interpretation is not conclusive and the data are not required for the interpretation of the AGO18 analysis.

2) We recommend a brief discussion of the follow points:

a) What is the proposed fate of the AGO18 bound miR168?

b) What is the sensing mechanism of viral RNA that triggers AGO18 expression?

---

## [Author Response]

[Editors’ note: the author responses to the first round of peer review follow.]

*All three reviewers have major but non-overlapping concerns about the manuscript. The major reservations are that*:

*1) The concept described overlaps with the previously described AGO1-AGO10 competition*.

We feel there is a major difference between our finding and previous findings on AGO1-AGO10 competition to make it novel and interesting. As discussed in the Results section, indeed there were previous examples of competitive binding of other miRNAs between other AGOs and AGO1 in *Arabidopsis*. This competition, however, does not affect AGO1 homeostasis. In contrast to these negative regulations of AGO1 activities, binding of miR168 by AGO18 in infected rice plants directly boosts AGO1 accumulation. Thus, among mechanisms of AGO-regulation of AGO1 homeostasis or activity, the up-regulation of AGO1 via AGO18 binding of miR168 represents a novel type of mechanism. This opens up the question of whether other AGOs function similarly in *Arabidopsis*, rice and other plants to up-regulate AGO1 or another AGO to impact developmental processes or defense responses.

*2) The study is incomplete. Some obvious experiments have not been done include (but are not restricted to) the use of miR168 resistant AGO1 mRNA and analysis of AGO2*.

We have performed additional experiments to demonstrate that expression of miR168-resistant *AGO1a* gene rescued the deficiency of *ago18* for viral resistance. The results are presented as a new section (see also the new Figure 7). We have performed new experiments to show that AGO18 specifically induces AGO1 but not AGO2 (Figure 4). Thus, AGO2 was not investigated further in current study. We have acknowledged the role of AGO2 in viral resistance in *Arabidopsis* in the text where appropriate.

*3) The fact that all RNAi lines (DCL1, DCL2, DCL3a, DCL3b, DCL4, AGO1) have the same phenotype, raising the possibility that there is interference of transgene RNAi and antiviral RNAi*.

To address this, we generated control transgenic rice plants that overexpress dsRNAs that target *pC2* (a glycoprotein) and *p4* (a viral disease related protein). In the *pC2* and *p4* RNAi plants, siRNAs accumulated at similar levels as those in the *dcl* RNAi lines (Figure 1—figure supplement 1). However, the production of these siRNAs had no effect on the infection of RSV as measured by the severity of disease symptoms or the accumulation of RSV (Figure 1—figure supplement 1). These data establish that elevated accumulation of RSV in each *dcl* RNA line resulted specifically from the down-regulated expression of DCL. How the rice DCL1 and DCL3 may have the novel antiviral function, in contrast to their counterparts in *Arabidopsis* and other dicot models, is an outstanding question to answer in future studies.

*Reviewer #1*:

*Overall these are nicely executed experiments although I would say that the obvious and most direct test of the hypothesis with a miR168 resistant AGO1 message is conspicuously absent*.

We have performed additional experiments to demonstrate that expression of miR168-resistant *AGO1a* gene rescued the deficiency of *ago18* for viral resistance. The results are presented as a new section in the main text (see also the new Figure 7).

*The data also do not test the prediction that AGO18 induction in virus-infected plants would mediate overexpression of AGO1 and that the timing of the effect would precede the maximal rate of virus accumulation*.

We indeed have data showing that infection by both RSV (Figure 2) and RDV (Figure 2—figure supplement 3) induced AGO18 expression, which was directly correlated with increased accumulations of *AGO1a* and *1b* mRNAs and proteins. Furthermore, over-expression of AGO18 in transgenic plants led to increased expression of *AGO1a* and *1b*, but had no effect on the expression of AGO2 (Figure 4). Thus, AGO18 can specifically up-regulate AGO1. The new data are added in the main text.

*The finding that the plant AGO-mediated antiviral silencing can be enhanced in virus-infected plants is not new and the decoy AGO mechanism is also not novel although it was described previously in a different context. Taking the sum of these considerations my view is that the paper in its present form is more suited to a specialist journal*.

We agree that the role of some AGOs in antiviral defense has been well demonstrated. However, our findings indicate that we are still far from understanding the mechanisms and that the field is still open for important new discoveries. We would like to discuss the novelty and importance of our work from three perspectives. First, to our knowledge, this is the first demonstration that the antiviral function of AGO1 requires the presence of another AGO. This represents an evolutionary novelty, at least in the monocots. Given the amplification and diversification of AGOs and other RNA silencing machinery components, our finding adds another important new layer of regulatory mechanism for studies on the evolution and mechanisms of RNA silencing machinery.

Second, the well-studied AGO1 and AGO2 are effectors of vsiRNAs and use their slicing activity in antiviral response. In contrast, AGO18 functions in a different mechanism – not through its slicing activity but through its small RNA binding activity. In the host defense field, this is a novel mechanism of AGO function.

Third, for the comment that “decoy AGO mechanism is also not novel”, we feel there is a major difference between our finding and previous findings to make it novel and interesting. As discussed, indeed there were previous examples of competitive binding of other miRNAs between other AGOs and AGO1 in *Arabidopsis*. This competition, however, does not affect AGO1 homeostasis. In contrast to these negative regulations of AGO1 activities, binding of miR168 by AGO18 in infected rice plants directly boosts AGO1 accumulation. Thus, among mechanisms of AGO-regulation of AGO1 homeostasis or activity, the up-regulation of AGO1 via AGO18 binding of miR168 represents a novel type of mechanism. This opens up the question of whether other AGOs function similarly in *Arabidopsis*, rice and other plants to up-regulate AGO1 or another AGO to impact developmental processes or defense responses.

*There are several interesting features in this paper that could be followed up to make a new paper that might be more appropriate to* eLife*. It would be interesting to know for example about the features of AGO18 that differentiate it from other AGO clades and to know more about the regulation of AGO18 in virus-infected plants. Specific questions would concern the role of insect vectors in the activation of this gene and the possibility (implied by*
Figure 2*) that there is a systemic signal associated with its induction. We know about signals induced in disease resistant plants but this would be a signal in susceptible plants. Is it an RNA or perhaps it is a JA response induced by the insect vectors of the virus?*

Our mock-inoculation experiments actually addressed this concern, which we should have explained in the original manuscript. Mock-inoculation with virus-free insect vectors had no effect on AGO18 expression, essentially eliminating the possibility that insect biting (and associated hormonal responses) in triggering AGO18 expression. We have discussed this in the main text.

*Reviewer #2*:

*In this manuscript, Wu et al., show that AGO18, a monocot-specific member of the ARGONAUTE family, is induced during virus infection and preferentially binds to miR168, a miRNA that normally down-regulates AGO1, which has antiviral RNAi activity. Thus, AGO18 induction titrates miR168, leading to up-regulation of AGO1 and increased resistance to viruses. The data are convincing and support the authors' conclusions*.

*However, the authors totally ignore the contribution of AGO2, both in the bibliographic introduction and in the development of their story. Indeed, at least five papers have shown that, in addition to AGO1, AGO2 contributes to antiviral RNAi activity (*[16]*;*
[51]*;*
[43]*;*
[20]*;*
[4]*). None of these papers is referenced in the Introduction and only a brief sentence in the Discussion says that, in addition to AGO1, AGO2 also has catalytic activity against viral RNA, but there is no reference cited*.

We have added and discuss the literature on AGO2 in the main text.

*The absence of investigation on AGO2 in this paper is a big concern, considering that the authors have previously shown that AGO2 is induced during virus infection, and actually more induced than AGO1 (*[11]*). Could AGO18 induction also lead to AGO2 up-regulation, and by which mechanism? This question needs to be addressed*.

We have performed new experiments to show that AGO18 specifically induces AGO1 but not AGO2 (Figure 4).

*In addition, the results obtained with the* dcl *RNAi lines are puzzling and actually raise a number of concerns regarding the RNAi strategy. Indeed, RNAi against either DCL1, DCL2, DCL3a, DCL3b or DCL4 increases infection rates. How is this possible, given the different role of each DCL? Isn't it that the production of very large amounts of dsRNA during RNAi somehow titrates the DCL(s) required for antiviral activity? If so, the results obtained during AGO1 RNAi are also questionable.*

We have addressed this concern by additional experiments and revised the text in the Results section as follows: “It should be noted that studies from *Arabidopsis* showed that DCL4 and DCL2 (in the absence of DCL4) are involved in antiviral defense (7). […] How the rice DCL1 and DCL3 may have the novel antiviral function, in contrast to their counterparts in *Arabidopsis* and other dicot models, is an outstanding question to answer in future studies.”

*Reviewer #3*:

*1) The authors should integrate their work in the literature context. Starting from the Introduction of well-studied AGO functions, the authors selectively emphasized their own work and ignored other elegant work (i.e. AGO7-miR390; Montgomery, et al.; 2008). Another example is that the mechanism presented here is essentially the same as the AGO10-AGO1 story (*[59]*). The only difference is AGO18-AGO1 for miR168 (which happens to feedback regulate AGO1). The way the authors wrote the manuscript suggests that their discovery is completely novel, if one does not go deep through the Discussion part*.

As we responded to a similar comment from reviewer #1 above, we have discussed our findings in the context of previous findings in the main text. We believe our finding adds a new and important layer regulatory mechanism of AGO functions that is of high and broad interest.

*2) The manuscript missed one piece of important information*: *could sequestering of miR168 under AGO18 promoter through miRNA sponge or target mimicry rescue ago18 responses to viral infection?*

We appreciate the reviewer’s great suggestion on the experiment to test whether AGO18 acts through miR168 regulation of AGO1 to confer viral resistance. We totally agree that results from the suggested experiment will be very informative. However, it takes more than one year to generate stable transgenic rice. Before we obtain these transgenic lines, we hope we can convince this reviewer that AGO18 plays its role in viral resistance through miR168 regulation of AGO1 by providing several lines of evidence: 1) AGO18 and AGO1 are both induced by RSV infection (Figure 2; Figure 4); 2) AGO18 competes with AGO1 for binding miR168 both in vivo (Figure 4) and in vitro (Figure 5); 3) upregulation of AGO1 by viral infection is compromised in *ago18* mutant; and 4) our new data showing that the expression of miR168-resistant *AGO1a* rescued the deficiency of *ago18* for viral resistance (Figure 7).

*3) In*
Figure 2
*it is apparent an increase in AGO18 levels upon RSV infection, especially in line #5*. *This is unexpected given that the gene is under a constitutive promoter. Is it possible that the virus infection affects the protein stability? It is known that AGO proteins are unstable in the apo form (this is without sRNA bound), could it be possible that virus miR168 induction regulates AGO18 stability? The authors could address this issue by checking transcript and protein levels of AGO18 in OE lines before and after virus infection. Is the concurrent increase of AGO18 and miR168 upon viral infection synergetic effect?*

We apologize for this confusion. We have repeated twice the Western blot experiments described in Figure 2 and realized that the lower AGO18 level in line #5 in the original figure was most likely due to inefficient transfer of protein to the membrane. We have replaced the figure with new Western blot results (Figure 2). As the reviewer suggested, we have also measured the mRNA and protein levels of AGO18 in OE lines before and after RSV infection by qRT-PCR and Western blot, respectively. We found that there was a mild increase in both AGO18 mRNA and protein levels in AGO18 OE lines upon RSV infection (see Figure 2 and Figure 8). Such increase is very likely attributed to the expression of endogenous AGO18 induced by RSV.Author response image 1.

*4) In the*
[Supplementary-material SD1-data]*, the highest accumulation of miRNA in AGO18 complexes is of miR528, which is more than 9-fold higher than in* AGO1b *or ∼3-fold* AGO1a *under virus infection. Given that miR528 is an important metabolic regulator during embryo development in rice, it seems very interesting to discuss its possible relationship to virus defense and whether AGO18 acts also as a decoy and not a silencing effector with this miRNA.*

We have been discussed this in the main text.

*5) In the Discussion part, the authors attempted to differentiate this finding from the AGO10-AGO1 weakens, instead of strengthening, their discussion, for having evidence of this phenomenon used not only for development but as an adaptive mechanism for viral defense in a distant organism as rice, makes it ubiquitous and conserved*.

As described above, we have revised our discussions to clarify how our findings add a new layer of regulatory mechanism to AGO functions.

[Editors' note: further revisions were requested prior to acceptance, as described below.]

We appreciate the favorable comments made by the Senior and Reviewing editors. In the revised manuscript, we have provided additional data or explanations to address some minor concerns raised by the two editors. We wish the revisions are sufficient and the manuscript is now acceptable for publication. Point-by-point responses are listed below.

*1) We strongly recommended removal of the* dcl *mutant analysis as the interpretation is not conclusive and the data are not required for the interpretation of the AGO18 analysis.*

As recommended, we have removed the data and text about the *dcl* mutant analyses.

*2) We recommend a brief discussion of the follow points*:

*a) What is the proposed fate of the AGO18 bound miR168*?

We have added the following paragraph in the Discussion: “Our findings suggest that AGO18-bound miR168 are not competent in cleaving *AGO1* mRNA in RSV-infected rice plants […]. It will be highly interesting to test these possibilities in future studies.”

*b) What is the sensing mechanism of viral RNA that triggers AGO18 expression*?

After submitting the manuscript, we obtained new data to address this question. We examined the possibility that viral proteins produced during infection induce AGO18 expression. To do this, we generated transgenic rice plants that over-express Myc-tagged RSV P2, CP, pC4 and P4 proteins, respectively. We found that AGO18 was induced only in the transgenic lines that over-expressed CP, but not in those that over-expressed three other viral proteins. This suggests that RSV CP is an effector to induce AGO18 expression during viral infection. These new results are included in a new supplemental figure (Figure 2—figure supplement 2).